# Cerebral Tissue Oxygen Saturation Is Enhanced in Patients following Transcatheter Aortic Valve Implantation: A Retrospective Study

**DOI:** 10.3390/jcm11071930

**Published:** 2022-03-30

**Authors:** Götz Schmidt, Hannes Kreissl, Ursula Vigelius-Rauch, Emmanuel Schneck, Fabian Edinger, Holger Nef, Andreas Böning, Michael Sander, Christian Koch

**Affiliations:** 1Department of Anesthesiology, Operative Intensive Care Medicine and Pain Therapy, Justus Liebig University Giessen, Rudolf-Buchheim-Strasse 7, 35392 Giessen, Germany; hannes.kreissl@med.uni-giessen.de (H.K.); ursula.vigelius-rauch@chiru.med.uni-giessen.de (U.V.-R.); emmanuel.schneck@chiru.med.uni-giessen.de (E.S.); fabian.edinger@chiru.med.uni-giessen.de (F.E.); michael.sander@chiru.med.uni-giessen.de (M.S.); christian.koch@chiru.med.uni-giessen.de (C.K.); 2Department of Cardiology and Angiology, Justus Liebig University Giessen, Klinikstrasse 33, 35392 Giessen, Germany; holger.nef@innere.med.uni-giessen.de; 3Department of Cardiac and Vascular Surgery, Justus Liebig University Giessen, Rudolf-Buchheim-Strasse 7, 35392 Giessen, Germany; andreas.boening@chiru.med.uni-giessen.de

**Keywords:** near-infrared spectroscopy, NIRS, TAVI, cerebral oximetry, cardiac surgery

## Abstract

Transcatheter aortic valve implantation (TAVI) has emerged as an alternative to surgical aortic valve replacement. The aim of this study was to evaluate whether a relevant alteration in cerebral tissue oxygen saturation (rSO_2_) could be detected following TAVI. Retrospective data analysis included 275 patients undergoing TAVI between October 2016 and December 2020. Overall, rSO_2_ significantly increased following TAVI (64.6 ± 10% vs. 68.1 ± 10%, *p* < 0.01). However, a significant rise was only observed in patients with a preoperative rSO_2_ < 60%. Of the hemodynamic confounders studied, hemoglobin, mean arterial pressure and blood pH were lowered, while central venous pressure and arterial partial pressure of carbon dioxide (PaCO_2_) were slightly elevated (PaCO_2_: 39 (36–43) mmHg vs. 42 (37–47) mmHg, *p* = 0.03; pH: 7.41 (7.3–7.4) vs. 7.36 (7.3–7.4), *p* < 0.01). Multivariate linear regression modeling identified only hemoglobin as a predictor of altered rSO_2_. Patients with a EuroScore II above 4% and an extended ICU stay were found to have lower rSO_2_, while no difference was observed in patients with postoperative delirium or between the implanted valve types. Further prospective studies that eliminate differences in potential confounding variables are necessary to confirm the rise in rSO_2_. Future research should provide more information on the value of cerebral oximetry for identifying high-risk patients who will require further clinical interventions in the setting of the TAVI procedure.

## 1. Introduction

Aortic valve stenosis is the most common form of degenerative heart valve disease, and is increasing in prevalence, particularly in ageing populations. When left untreated, impaired left ventricular outflow increases afterload, thus generating left-ventricular pressure overload and leading to concentric hypertrophy and subsequent heart failure [1]. Due to technological and procedural advances, transcatheter aortic valve implantation (TAVI) has emerged as an attractive alternative to surgical aortic valve replacement (SAVR), as it does not require sternotomy and extracorporeal circulation [2]. Compared to SAVR, TAVI is associated with lower mortality, shorter length of hospital stays, and a more rapid return to normal life [3,4,5,6]. In contrast, patients undergoing SAVR have a lower risk of paravalvular regurgitation, permanent pacemaker implantation, and need for secondary valve intervention [7,8,9]. Thus, according to current European and American guidelines for the management of valvular heart disease, TAVI is favored in frail patients who are aged over 75 or 80 years, or who have a reduced life expectancy or a high surgical risk [10,11]. Currently, however, the indications for TAVI are expanding beyond otherwise inoperable patients [2]. Evidence is emerging that patients of intermediate surgical risk aged between 65 and 75 years old may also benefit from TAVI [12]. As such, current guidelines on the management of valvular heart disease recommend that these patients should be considered by interdisciplinary heart valve teams, and their clinical, anatomical, and procedural factors should be weighed for an individual approach [10,11]. Although TAVI is less invasive, heart valve interventions in elderly and often frail patients represent a challenge for perioperative anesthetic management. During the TAVI procedure, balloon valvuloplasty, the implantation of a balloon-expandable valve and post-dilatation are performed during functional circulatory arrest induced by rapid ventricular pacing (RVP) [13]. The obstruction of the left ventricular outflow tract during valvuloplasty, followed by severe aortic regurgitation, can also lead to hemodynamic deterioration through sudden left ventricular volume overload. The option to establish a cardiopulmonary bypass, or even switch to SAVR in the event of interventional disturbances, must therefore be guaranteed at any time during the procedure [14]. Consequently, anesthetic care is performed according to the standards of conventional cardiac surgery and includes cerebral monitoring. While the bispectral index using processed electroencephalogram is utilized to assess the depth of anesthesia, cerebral near-infrared spectroscopy (NIRS) can be used to measure regional cerebral tissue oxygen saturation (rSO_2_) by transmitting near-infrared light to a sensor attached to the patient’s forehead above the frontal lobe [15]. As oxygenated and deoxygenated hemoglobin have different absorption spectra, the rSO_2_ can be calculated using Lambert-Beer law. Near-infrared light passes through approximately 70–75% venous and 25–30% arterial blood volume; thus, displayed values reflect primarily venous oxygen consumption as a representation of regional cerebral blood flow and global cardiopulmonary function [15,16]. Depending on the device used, a rSO_2_ between 60 and 70% at room air is considered normal in cardiac surgical patients, while a lower baseline rSO_2_ is associated with higher incidence of postoperative delirium and mortality [17,18,19]. Although current data appear to be inconclusive, evidence is emerging that cerebral oximetry measurement can improve neurological outcomes in patients by reducing the incidence and severity of postoperative cognitive disorder (POCD) and delirium [20,21,22]. During TAVI, hemodynamic condition is altered promptly due to regained unlimited left-ventricular outflow. It therefore follows that restored cardiac physiology after TAVI may improve cerebral blood flow and thus improves cerebral oxygen consumption. Consequently, the aim of this study was to evaluate whether a relevant alteration in rSO_2_ could be detected following TAVI.

## 2. Materials and Methods

Study design. This retrospective, single-center cohort study was approved by the local ethics committee of the medical faculty of the Justus-Liebig-University, Giessen, Germany (AZ 243/20). According to the institutional standard of care, cerebral oximetry has been routinely used during TAVI procedures since October 2016. Whenever possible, TAVI was performed under conscious sedation with additional ilioinguinal nerve block using remifentanil (0.03–0.05 µg/kg/min) and propofol (1–1.5 mg/kg/h), targeting a bispectral index > 70. Otherwise, general anesthesia with remifentanil (0.1–0.2 µg/kg/min) and propofol (2–3 mg/kg/h) was used. When clinically indicated, midazolam (1 mg intravenously) was given as premedication after the arrival of the patient in the hybrid operating room. Clinical data records were screened retrospectively for patients who underwent TAVI at the university hospital of Giessen from October 2016 to December 2020 (Operation and Procedure Classification System, codes 5-35a.0, 5-35a.01–5-35a.04). Patients with sufficient availability of records on rSO_2_ and clinical information that met the baseline and procedural characteristics were included in the analysis. Sufficient available rSO_2_ data was defined as at least one value recorded during a five-minute period at the beginning of the intervention, and at least one value recorded during a five-minute period at the end. At that time, patients had to be under stable general anesthesia or conscious sedation that was maintained throughout the procedure. Patients who did not meet these criteria were excluded, as were those who underwent failed interventions.

Data acquisition. After eligible patients were identified, anesthetic protocols, clinical data records and diagnostic results were reviewed manually. Baseline characteristics included age, sex, body mass index, cardiovascular risk factors and relevant comorbidities such as peripheral artery disease, carotid artery stenosis, prior stroke or transient ischemic attack, chronic obstructive pulmonary disease, pulmonary hypertension, and chronic kidney failure. EuroScore II, pre- and post-operative electrocardiogram, echocardiography, and chest X-ray findings were recorded, as well as intraprocedural data, including anesthetic management. Cerebral oximetry was measured using an INVOS 5100C Cerebral/Somatic Oximeter (Medtronic, Minneapolis, MN, USA). Until May 2020, cerebral oximetry data were manually documented in the digital anesthetic protocol at the discretion of the anesthesiologist. After May 2020, oximetry data was recorded automatically every 5 min. Values from both cerebral hemispheres were averaged, unless the absolute difference between them was not greater than 10%; otherwise, they were registered separately. The rSO_2_ values were contextualized with pulmonary and hemodynamic parameters to determine any relevant influence besides the procedural impact. To evaluate the comparability of each pair of cerebral oximetry values, hemoglobin (Hb), mean arterial pressure (MAP), central venous pressure (CVP), inspirational fraction of oxygen (FiO_2_), arterial partial pressure of oxygen (PaO_2_) and carbon dioxide (PaCO_2_), arterial oxygen saturation (SaO_2_) and blood pH were documented for each oximetry value, and were analyzed as potential confounding variables. Outcome analysis included conversion to open surgery, the necessity for cardiopulmonary resuscitation (CPR), presence of POCD or POD according to the Confusion Assessment Method for Intensive Care Unit (CAM-ICU) criteria, and death [23]. Length of intensive care unit (ICU) and hospital stay were calculated from the administrative records. When available, EuroScore II and mortality rates were obtained from a local registry of patients undergoing TAVI (Kerckhoff Biomarkerregister (BioReg), Bad Nauheim, Germany); otherwise, the EuroScore II was calculated retrospectively, and clinical records were used to evaluate in-hospital mortality [24].

CPR, intraoperative and in-hospital death, and conversion to open surgery were included in a composite endpoint summarizing adverse events. Cerebral saturations and prevalence of catecholamine administration (summarized as administration of noradrenaline or adrenaline) in clinically relevant subgroups were compared with their respective complementary groups. These subgroups included individuals with a higher preoperative EuroScore II, POCD and POD, extended ICU or hospital stay, and adverse events. According to European guidelines, if there are no other clinical implications, SAVR is favored in patients with a EuroScore II under 4% [10]. A high EuroScore II group was therefore defined as a EuroScore II above 4%. Extended ICU and hospital stay were defined as any duration above the median length of stay.

Statistical analysis. Categorial variables are presented as numbers and percentages. Chi-squared test or the Fisher’s exact test was used to compare categorical data. Normally distributed continuous variables are presented as mean ± standard deviation (SD), while non-normally distributed variables are described by the median and interquartile range (Q1–Q3). Cerebral oximetry, confounding parameters, and patient subgroups were compared using a one-way ANOVA followed by Tukey’s test. In addition, a multivariate linear regression model was used to evaluate the influence of confounding parameters on changes in cerebral oximetry. Differences in pre- and post-operative confounding parameters were calculated and compared to their corresponding alterations in cerebral oximetry. Moreover, 2-tailed values of *p* < 0.05 were considered statistically significant.

## 3. Results

### 3.1. Patient Characteristics

A total of 405 patients were identified who underwent TAVI between October 2016 and December 2020. After screening for eligibility, 275 patients were included in the final analysis (Figure 1).

Baseline and procedural characteristics. Patient characteristics, baseline echocardiography, perioperative anesthetic management and procedural details are summarized in Table 1. The median age was 81 (77–84) years and patients were noted to have a high prevalence of cardiopulmonary comorbidities. The mean EuroScore II was 6.43 ± 5.7%, and baseline echocardiography findings showed severe aortic stenosis with an aortic valve area of 0.7 (0.6–0.9) cm^2^ (*n* = 239), a peak pressure gradient of 66 ± 22 mmHg (*n* = 255), a mean pressure gradient of 40 ± 14 mmHg (*n* = 259), and a peak velocity of 4.9 ± 7.24 m/s^2^ (*n* = 239). Left ventricular ejection fraction was mostly preserved (LVEF < 30%: 5.2%, *n* = 237). Overall, most procedures were carried out under conscious sedation (90%), and there was a low conversion rate to general anesthesia (5.1%). Perioperative fluid therapy was provided using mainly crystalloids (1000 (647–1071) mL), while the use of blood products and colloids was negligible. Most patients required intraoperative catecholamine administration (74.9%); however, the median cumulative dose was low (noradrenaline: 59 (2–169) µg, adrenaline: 0 (0–0) µg). A transfemoral approach for TAVI was considered standard procedure when not contraindicated and was used for 97.1% of the interventions. 81.1% of the implanted valves were self-expandable valves that were largely implanted following balloon valvuloplasty (71.3%) during RVP, while 18.9% were balloon-expanded valves. The need for post-dilatation requiring repeat RVP was 13.8%. Second valve implantation (“TAVI in TAVI”) was performed in 4 cases (1.5%) and 2 patients required intraoperative electrical cardioversion (0.7%).

### 3.2. Perioperative Outcome

The overall rate of intraoperative adverse events was low (Table 2). None of the procedures required switching to SAVR. Intraoperative cardiopulmonary resuscitation had to be performed in 10 patients (3.6%), and new left bundle branch block was detected in 33 patients (12.0%). Temporary or permanent pacemaker dependency occurred in 10 (3.6%) and 33 (12.0%) patients, respectively, and this was largely due to new high-grade atrioventricular block (10.9%). Follow up echocardiography revealed high procedural success (Table 3), though some degree of paravalvular leakage was found in 113 patients (42.5%). Median length of ICU and hospital stay was 4 (2.2–6.7) days and 11 (8.5–17.0) days, respectively. POCD or POD occurred in 42 patients (15.3%) and 9 patients died during the index hospital stay (3.3%).

### 3.3. Cerebral Oximetry

Overall, median alteration of rSO_2_ was +4.0 (−1.0–+8.0) %, and absolute values of rSO_2_ significantly increased from 64.6 ± 10% to 68.1 ± 10% following TAVI (*p* < 0.01; Figure 2a). Stratified by preoperative rSO_2_ values, cerebral oximetry was significantly increased in patients with a preoperative rSO_2_ that was lower than 50% (45.1 ± 3.7% vs. 53.6 ± 11.1%, *p* < 0.01) and between 51–60% (56.6 ± 3.0% vs. 63.2 ± 8.5%, *p* < 0.001). No significant changes were observed in patients with a preoperative rSO_2_ between 61–70% (65.7 ± 2.9% vs. 68.0 ± 6,4%, *p* = 0.16) and above 70% (76.8 ± 5.2% vs. 77.0 ± 7.22%), *p* = 1.0; Figure 2b).

### 3.4. Hemodynamic Confounder

The changes observed in hemodynamic parameters are illustrated in Table 3. Hb (11.6 ± 1.8 g/dL vs. 10.3 ± 1.7 g/dL, *p* < 0.01) and MAP (91 ± 17 mmHg vs. 76 ± 18 mmHg, *p* < 0.01) were found to significantly decrease postoperatively, while CVP increased during the procedure (11 ± 7 mmHg vs. 13 ± 7 mmHg, *p* < 0.01). PaCO_2_ also increased postoperatively (39 (36–43) mmHg vs. 42 (37–47) mmHg, *p* = 0.03), and this was accompanied by reciprocal changes in pH (7.41 (7.3–7.4) vs. 7.36 (7.3–7.4), *p* < 0.01). No significant changes were observed in PaO_2_ (128 (98–169) mmHg vs. 125 (90–160) mmHg, *p* = 0.49) and SaO_2_ (99 (98–99) % vs. 99 (97–99) %, *p* = 0.18).

To evaluate the influence of hemodynamic confounders on cerebral rS_O2_, differences in the pre- and post-operative variables were subsequently calculated and assessed against the changes in rSO_2_ using a multivariate linear regression model. The result of this analysis showed that only changes in Hb were independent predictors of altered cerebral rS_O2_ (*p* < 0.01; *n* = 60; adjusted R^2^ = 0.17, overall *p* = 0.029). According to the model, variations in MAP, CVP, pH, PaCO_2_, PaSO_2_ and SaO_2_ were of no predictive value for rSO_2_ in this study (Table 4).

### 3.5. Subgroup Analysis

rS_O2_ and catecholamine administration were comparable among patients with self-expanding or balloon-expandable valves and patients with POCD or POD following TAVI (Table 5). However, patients with a EuroScore II above 4% and those who had longer ICU or hospital stays had lower preoperative (63.5 ± 10.1% vs. 66.2 ± 9.7%, *p* = 0.03; 62.6 ± 10.0% vs. 66.2 ± 9.7%, *p* < 0.01; 61.2 ± 9.2% vs. 67.5 ± 9.8%, *p* < 0.01) and postoperative rSO_2_ (66.8 ± 10.9% vs. 69.8 ± 8.5%, *p* = 0.01; 66.5 ± 10.0% vs. 69.7 ± 9.9%, *p* < 0.01; 64.8 ± 10.4% vs. 70.9 ± 8.9%, *p* < 0.01), and had higher catecholamine requirements postoperatively (37% vs. 25%, *p* = 0.03; 40.4% vs. 24.5%, *p* < 0.01; 40.8% vs. 25.3%, *p* < 0.01). In contrast, preoperative catecholamine administration was comparable among these groups. Patients who experienced adverse events showed comparable preoperative rSO_2_ values (64.9 ± 10.1% vs. 60.9 ± 7.2%, *p* = 0.14), but had higher preoperative catecholamines requirements (26.7% vs. 7.7%, *p* = 0.03). Postoperative rSO_2_ values were lower (68.4 ± 10.1% vs. 62.5 ± 8.4%, *p* = 0.03), and were accompanied by a higher catecholamine requirement (29.2% vs. 86.7%, *p* < 0.01). In contrast, the numerical rise in rSO_2_ was comparable between all of the analyzed subgroups.

## 4. Discussion

A significant rise in cerebral rS_O2_ was observed in this retrospective cohort following TAVI. As the new competent aortic valve restores adequate cardiac output and oxygen delivery, this rise could be attributed to improved hemodynamic function. However, although significant, the actual numerical rise in rSO_2_ was small in our cohort, and it must therefore be considered that confounding factors contributed to these slightly elevated rSO_2_ values. To date, evidence on the course of cerebral NIRS following TAVI has focused mainly on the episode of RVP [25,26,27]. Consistent to the functionally circulatory arrest that is achieved during RVP, rSO_2_ was seen to fall significantly. Interestingly, a decrease of more than 20% resulted in a higher incidence of POD, and these patients showed also lower rSO_2_ baseline values [26]. Desaturation after RVP was also followed by transient hyperemia leading to elevated rSO_2_. However, rSO_2_ returned to its baseline value only a few minutes after RVP, and the rise in rSO_2_ observed after the procedure in this study should not therefore be attributed to recent RVP [27]. Thus, hyperemia could be attributed to acid metabolites accumulated during RVP leading to temporarily enhanced cerebral blood flow. Temporarily impaired cerebral autoregulation after prompt alteration of cardiac hemodynamic following TAVI may also be considered as explanation for the temporary rise in rSO_2_ in our study. In other clinical settings, such as in carotid endarterectomy procedures, cerebral hyperperfusion accompanied by neurological deficits is a well-known clinical syndrome that is associated with highly elevated rSO_2_ values [28]. It should be noted, however, that the magnitude of the rise in rSO_2_ observed in this study was much lower than in those patients. Furthermore, although values of baseline rSO_2_ were comparable, Suppan et al. observed improved rSO_2_ and cardiac output following cardiopulmonary exercise testing 5 days after TAVI [29]. Consequently, the rise in rSO_2_ observed in this study might not be a short-term change.

The increase in rSO_2_ in this study substantially differed between subgroups, depending on their preoperative cerebral oximetry values. The rise observed in patients with low preoperative rSO_2_ (<50%, 51–60%) was significantly greater than in patients with a preoperative rSO_2_ > 60%, whose values were unchanged postoperatively. This could suggest that patients with aortic valve stenosis and a preoperative rSO_2_ < 60% even suffer from hemodynamic restrictions at rest due to aortic valve stenosis, and this manifests not only as clinical symptoms, but also as imminent cardiac decompensation. In general, rSO_2_ values are subject to large variations between individuals [30]. In their study, Fanning et al. were unable to detect a difference between pre- and post-operative values in patients undergoing TAVI with general anesthesia, and similar results were found by Mayr et al., who compared rSO_2_ during conscious sedation with rSO_2_ during general anesthesia [27,31]. Although these results were not statistically investigated, a numerical increase in rSO_2_ was observed at the end of the TAVI procedure in both general anesthesia and conscious sedation cohorts. However, it should be noted that both trials utilized only small study cohorts, and the comparison of rSO_2_ during conscious sedation and general anesthesia may still be affected by anesthetic care. In our study, Hb and MAP were found to be significantly reduced, while CVP was raised postoperatively. However, these alterations were not found to be associated with an increase in rSO_2_: MAP remained within the autoregulatory range in our study cohort. The association between low Hb and lower rSO_2_ has been shown in cardiac surgery patients [32]. Thus, the reduction in Hb seen in this study results in a concordant reduction in oxygen supply, which would therefore reduce rSO_2_, rather than raise it [28,33]. Cerebral perfusion pressure and cerebral blood flow are reduced by higher CVP; hence increased venous filling or congestion illustrated by elevated CVP would be expected to lower rSO_2_, not raise it. However, PaCO_2_ was slightly, but still significantly, raised in our cohort, and consecutive alterations in pH were also detected. The numerical change in these values was lower than those observed by Mayr et al.; nevertheless, alterations in PaCO_2_ must be analyzed with particular care [25]. In contrast to other hemodynamic parameters, there is no cerebral autoregulation mechanism for PaCO_2_. This means that, within physiological ranges, cerebral blood flow enhances linearly with rising PaCO_2_ [34,35]. The relation of PaCO_2_ and rSO_2_ has been examined in various clinical settings using wide ranges of target PaCO_2_, from 35 mmHg to 55 mmHg [36,37,38,39]. rSO_2_ increases in association with PaCO_2_, but the magnitude of this increase is not fully understood. For example, during cardiopulmonary bypass, rSO_2_ increased from 55% to 64% after an increase in PaCO_2_ from 38 mmHg to 52 mmHg [39]. In another study by Sørensen et al. investigating changes in end tidal CO_2_ and rSO_2_ during reperfusion following aortic arch surgery, an increase of 3.75 mmHg was observed in etCO_2_, followed by a 2% increase in rSO_2_ [38]. In their study, Wong et al. randomly assigned patients undergoing major non-cardiac surgery to median target PaCO_2_ values of either 34.8 mmHg and 51.5 mmHg and detected a +19% increase in rSO_2_ in the hypercapnia group [37]. Compared to these trials, the rise observed in PaCO_2_ in this study from median 39 (36–43) mmHg to median 42 (37–47) mmHg was very low, but the rise in rSO_2_ was greater than that observed by Sørensen et al. [38]. These findings suggest that the observed rise in rSO_2_ in our cohort might be influenced by factors other than changes in PaCO_2_. This is supported by the result of our multivariate linear regression model, which found that altered Hb alone was a predictor of altered rSO_2_ in our cohort. It should be noted, that changes in PaCO_2_ are accompanied by corresponding changes in pH, which facilitates the oxygen release from hemoglobin via the Bohr effect. The increase in PaCO_2_ is likely to be influenced by the anesthetic procedure. Most of the patients in this study underwent TAVI with conscious sedation using propofol and remifentanil. In a randomized controlled trial by Mayr et al., patients under conscious sedation with propofol and opioid were shown to have significantly elevated levels of PaCO_2_ after induction and during valvuloplasty compared to patients under general anesthesia [25]. Hypoventilation is affected not only by the anesthetic procedure, but also by the choice of drugs used for sedation. Compared to propofol and opioids, conscious sedation with dexmedetomidine results in a lower incidence of hypercapnia and an overall reduction in PaCO_2_ [31]. However, as shown in this study, PaCO_2_ levels remained consistently within clinically safe ranges, regardless of the choice of sedative.

In our subgroup analysis, rSO_2_ and administration of catecholamines were comparable between patients treated with self-expandable and balloon-expanded valves, and these results are consistent with those of Eertmans et al. [40]. Comparable values of rSO_2_ were observed pre- and post-operatively, but intraoperative desaturation was more substantial during the implantation of balloon-expandable devices, which is due to the implantation technique. More substantial desaturations during RVP were observed in patients with a higher EuroScore, who also showed an overall reduction in rSO_2_ in our cohort [26]. Preoperative rSO_2_ was lower in patients with a higher EuroScore who underwent on-pump cardiac surgery, and these patients were found to have increased postoperative mortality [18,41]. Lower rSO_2_ was also shown to be an independent predictor of POD following on-pump cardiac surgery [41]. Whether this observation also applies following TAVI is still unclear, however, in our cohort, there was no difference between rSO_2_ and catecholamine requirement in patients with POD or POCD. This finding could be attributed to a shorter procedural duration, lower usage of general anesthesia, and reduced invasiveness of the procedure that avoided cardiopulmonary bypass, which is associated with a higher incidence of POD [42]. Nevertheless, patients with longer ICU and hospital stay were found to have lower preoperative rSO_2_ values in our cohort. This may be attributed to stronger restrictions of cardiopulmonary function and comorbidities. Interestingly, the intraoperative increase in rSO_2_ was comparable, but postoperative rSO_2_ remained significantly lower. Cerebral oximetry of these patients may therefore indicate a successful procedure, but oximetry values remained at an overall lower level than in patients who had shorter stays in the ICU and hospital. That a higher postoperative catecholamine requirement was observed both in patients with longer ICU and hospital stays and in those with a higher EuroScore II further suggests that these results indicate more difficulties in adapting to new cardiovascular conditions. Similarly, patients who experienced adverse events had lower postoperative rSO_2_ values and higher catecholamine requirements. However, despite comparable preoperative rSO_2_, their preoperative catecholamine requirements were already higher compared to those patients who did not experience adverse events. This suggests that intraoperative disturbances are not the only factors leading to severe circulatory restrictions represented by lower rSO_2_ values.

Of course, several limitations of our study must be acknowledged. The first is that retrospective data were used for analysis. For example, incidence of POD was assessed retrospectively using CAM-ICU criteria and therefore relevant under-detection, particularly of hypoactive delirium, cannot be ruled out. Second, problems with availability made it impossible to analyze cerebral oximetry data without the presence of additional oxygen administration. Cerebral oximetry during general anesthesia or conscious sedation was considered, and the measurements may therefore still have been affected by anesthetic care. Although cerebral oximetry values were measured continuously throughout the procedure, they were only partly documented, either at the anesthesiologist’s discretion, or automatically every 5 min. Additionally, it should be noted that the statistical power of some subgroup analysis may have been impaired due to small group sizes.

## 5. Conclusions

Our study shows a significant rise in rSO_2_ following TAVI. However, the actual numerical increase observed was small, and levels of PaCO_2_, a potential powerful confounder, also increased. Nevertheless, we showed that patients from clinically relevant subgroups with a higher EuroScore II, extended hospital or ICU stay, and adverse events had lower rSO_2_ levels and higher catecholamines requirements. Further prospective studies that eliminate differences in potential confounding variables are necessary to confirm the rise in rSO_2_ following TAVI. A synchronous comparison of different sedation regimes, including the use of dexmedetomidine, may be helpful to rule out the role of sedative-induced hypoventilation. Thus, future research may provide more information on the value of cerebral oximetry for identifying high-risk patients who will require further clinical interventions in the setting of the TAVI procedure.

## Figures and Tables

**Figure 1 jcm-11-01930-f001:**
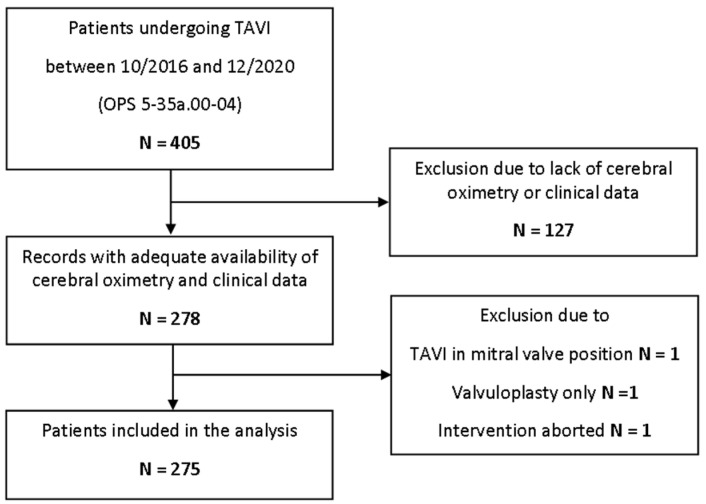
Study flowchart. TAVI = transcatheter aortic valve implantation.

**Figure 2 jcm-11-01930-f002:**
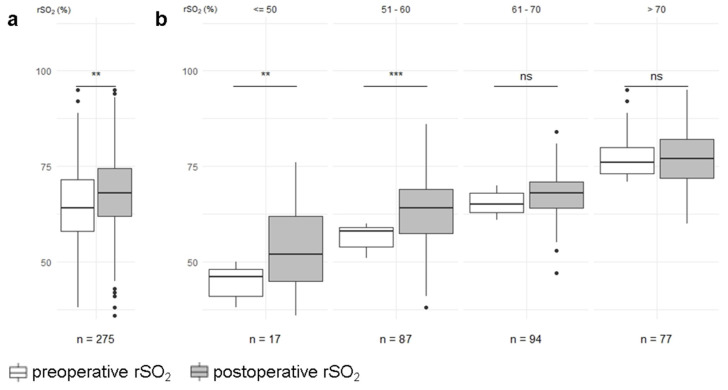
(**a**) Cerebral rSO_2_ was enhanced after TAVI. (**b**) Analysis stratified by preoperative values revealed a significant increase only in patients with preoperative rSO_2_ ≤ 50% and 51–60%. **: *p* < 0.01; ***: *p* < 0.001; ns: not significant; rSO_2_ = regional cerebral oxygen saturation.

**Table 1 jcm-11-01930-t001:** Patient characteristics, anesthetic management, and procedural details.

Characteristics	*n* = 275
**Patient characteristics**	
male-no. (%)	139 (50.5)
median age (Q1–Q3)-years	81 (77–84)
mean Body-Mass-Index (SD)-kg/m^2^	27 (±4)
arterial hypertension-no. (%)	230 (83.6)
pulmonary hypertension-no. (%)	14 (5.1)
coronary artery disease-no. (%)	190 (69.1)
peripheral vessel disease-no. (%)	34 (12.4)
carotid artery stenosis-no. (%)	45 (16.4)
prior stroke or transient ischemic attack-no. (%)	29 (10.5)
chronic kidney disease-no. (%)	69 (25.1)
chronic obstructive pulmonary disease-no. (%)	35 (12.7)
current smoker-no. (%)	15 (5.5)
family history of CAD-no. (%)	24 (8.7)
pacemaker-no. (%)	31 (11.3)
pleural effusion present-no. (%)	57 (22.2)
EuroScore II (SD)-%	6.43 (±5.7)
**Anesthetic management**	
conscious sedation-no. (%)	239 (86.9)
general anesthesia-no. (%)	22 (8.0)
switch to general anesthesia-no. (%)	14 (5.1)
amount of crystalloids (Q1–Q3)-mL	1000 (647–1071)
amount of colloids (Q1–Q3)-mL	0 (0–0)
amount of red blood cell concentrate (Q1–Q3)-mL	0 (0–0)
amount of fresh frozen plasma (Q1–Q3)-mL	0 (0–0)
catecholamine administration-no. (%)	206 (74.9)
cumulative dosage of noradrenaline (Q1–Q3)-µg	59 (2–169)
cumulative dosage of adrenaline (Q1–Q3)-µg	0 (0–0)
duration of anesthesia (SD)-min	131 (±36)
**Procedural details**	
duration of intervention (Q1–Q3)-min	55 (45–69)
transfemoral approach-no. (%)	267 (97.1)
transapical approach-no. (%)	5 (1.8)
subclavian approach-no. (%)	3 (1.1)
balloon-expandable valve-no. (%)	52 (18.9)
self-expandable valve-no. (%)	223 (81.1)
valvuloplasty-no. (%)	196 (71.3)
postdilatation-no. (%)	38 (13.8)
Second valve implantation-no. (%)	4 (1.5)
number of rapid pacing (Q1–Q3)-no.	1 (1–2)
number of fast pacing (Q1–Q3)-no.	1 (0–1)
Cardioversion-no. (%)	2 (0.7)

**Table 2 jcm-11-01930-t002:** Adverse events, postprocedural ECG and outcome.

Characteristics	*n* = 275
**Postprocedural ECG**	
new high-grade AVB-no. (%)	30 (10.9)
new LBBB-no. (%)	45 (16.9)
new pacemaker dependency, permanent-no. (%)	33 (12.0)
new pacemaker dependency, temporary-no. (%)	10 (3.6)
**Outcome**	
Any paravalvular leak-no. (%)	113 (42.5)
Median length of ICU stay (Q1–Q3)-days	4 (2.2–6.7)
Median length of hospital stay (Q1–Q3)-days	11 (8.5–17.0)
POCD or delirium-no. (%)	42 (15.3)
**Adverse events**	
Conversion to open surgery-no. (%)	0 (0.0)
CPR-no. (%)	10 (3.6)
Intraoperative mortality-no. (%)	0 (0.0)
In-hospital mortality-no. (%)	9 (3.3)

**Table 3 jcm-11-01930-t003:** Overall rSO_2_, hemodynamic confounders and follow-up echocardiography.

Characteristics	*n*	Preoperative	Postoperative	*n*	*p*
**Cerebral oximetry**					
ΔrSO_2_ (Q1–Q3)-%	275	4.0 (−1.0–8.0)		275	
rSO_2_ (SD)-%	275	64.6 (±10)	68.1 (±10)	275	<0.01
**Confounding parameters**					
Hb (SD)-mg/dL	262	11.6 (±1.8)	10.3 (±1.7)	213	**<0.01**
MAP (SD)-mmHg	275	91 (±17)	76 (±18)	275	**<0.01**
CVP (SD)-mmHg	205	11 (±7)	13 (±7)	255	**<0.01**
pH (Q1–Q3)-no.	101	7.41 (7.3–7.4)	7.36 (7.3–7.4)	121	**<0.01**
P_a_CO_2_ (Q1–Q3)-mmHg	101	39 (36–43)	42 (37–47)	121	**0.03**
P_a_O_2_ (Q1–Q3)-mmHg	101	128 (98–169)	125 (90–160)	119	0.49
S_a_O_2_ (Q1–Q3)-%	101	99 (98–99)	99 (97–99)	119	0.18
**Echocardiography**					
LVEF (SD)-%	237	54 (±11)	55 (±9)	176	0.38
AV-PPG (SD)-mmHg	255	66 (±22)	16 (±8)	237	**<0.01**
AV-MPG (SD)-mmHg	259	40 (±14)	9 (±4)	225	**<0.01**

rSO_2_ = regional cerebral oxygen saturation; Hb = hemoglobin; MAP = mean arterial pressure; CVP = central venous pressure; FiO_2_ = inspirational oxygen fraction; PaO_2_ = arterial partial pressure of oxygen; PaCO_2_ = arterial partial pressure of carbon dioxide; SaO_2_ = arterial oxygen saturation; SpO_2_ = partial oxygen saturation; LVEF = left ventricular ejection fraction; AV = aortic valve; PPG = peak pressure gradient; MPG = mean pressure gradient.

**Table 4 jcm-11-01930-t004:** Multivariate linear regression model of the alteration of rSO_2_ and differences in confounders only revealed Hemoglobin as predictor of altered rSO_2_.

Variable	Estimate	Standard Error	*p*
**ΔrSO_2_**			
Intercerpt	5.06	1.51	**<0.01**
ΔHb	4.21	1.12	**<0.01**
ΔMAP	−0.03	0.04	0.48
ΔCVP	−0.06	0.15	0.70
ΔpH	−23.98	31.71	0.45
ΔP_a_CO_2_	0.10	0.24	0.66
ΔP_a_O_2_	0.00	0.02	0.78
ΔS_a_O_2_	0.27	0.35	0.44
*n* = 60	Adjusted R^2^: 0.17; ***p* = 0.029**		

rSO_2_ = regional cerebral oxygen saturation; Hb = hemoglobin; MAP = mean arterial pressure; CVP = central venous pressure; FiO_2_ = inspirational oxygen fraction; PaO_2_ = arterial partial pressure of oxygen; PaCO_2_ = arterial partial pressure of carbon dioxide; SaO_2_ = arterial oxygen saturation; SpO_2_ = partial oxygen saturation; LVEF = left ventricular ejection fraction.

**Table 5 jcm-11-01930-t005:** Subgroup analysis revealed lower rSO_2_ and higher need for catecholamines in patients with higher EuroScore II, longer ICU and hospital stay.

Characteristics			*p*
**Valve Type**	**Self-Expanding** ***n* = 223**	**Balloon-Expandable** ***n* = 52**	
ΔrSO_2_ (Q1–Q3)-%	3.0 (−1.0–8.0)	4.0 (−1.0–8.0)	0.98
preoperative rSO_2_ (SD)-%	64.7 (±10.3)	64.2 (±8.8)	0.71
catecholamine administration-no. (%)	18 (8.1)	6 (11.5)	0.42
postoperative rSO_2_ (SD)-%	68.2 (±10.2)	67.7 (±9.8)	0.73
catecholamine administration-no. (%)	66 (29.6)	23 (44.2)	0.05
**POCD or Delirium**	**no** ***n* = 233**	**yes** ***n* = 42**	
ΔrSO_2_ (Q1–Q3)-%	3.0 (−1.0–8.0)	4.0 (0–7.75)	0.97
preoperative rSO_2_ (SD)-%	64.9 (±10.3)	63.2 (±8.0)	0.33
catecholamine administration-no. (%)	21 (9.0)	3 (7.1)	1.00
postoperative rSO_2_ (SD)-%	68.3 (±10.5)	66.7 (±7.5)	0.34
catecholamine administration-no. (%)	71 (30.5)	18 (42.9)	0.15
**EuroScore II**	**≤4%** ***n* = 116**	**>4%** ***n* = 151**	
ΔrSO_2_ (Q1–Q3)-%	4.0 (−1.0–8.0)	3.0 (−1.0–8.0)	0.72
preoperative rSO_2_ (SD)-%	66.2 (±9.7)	63.5 (±10.1)	**0.03**
catecholamine administration-no. (%)	9 (7.8)	15 (9.4)	0.67
postoperative rSO_2_ (SD)-%	69.8 (±8.5)	66.8 (±10.9)	**0.01**
catecholamine administration-no. (%)	29 (25.0)	60 (37.7)	**0.03**
**length of ICU stay**	**short (≤4 d)** ***n* = 146**	**long (>4 d)** ***n* = 129**	
ΔrSO_2_ (Q1–Q3)-%	3.0 (−1.0–7.0)	4.0 (−1.0–8.3)	0.38
preoperative rSO_2_ (SD)-%	66.2 (±9.7)	62.6 (±10.0)	**<0.01**
catecholamine administration-no. (%)	9 (6.5)	15 (11.0)	0.20
postoperative rSO_2_ (SD)-%	69.7 (±9.9)	66.5 (±10.0)	**<0.01**
catecholamine administration-no. (%)	34 (24.5)	55 (40.4)	**<0.01**
**length of hospital stay**	**short (≤11 d)** ***n* = 150**	**long (>11 d)** ***n* = 125**	
ΔrSO_2_ (Q1–Q3)-%	3.5 (−1.0–7.75)	4.0 (−1.0–8.0)	0.77
preoperative rSO_2_ (SD)-%	67.5 (±9.8)	61.2 (±9.2)	**<0.01**
catecholamine administration-no. (%)	10 (6.7)	14 (11.2)	0.20
postoperative rSO_2_ (SD)-%	70.9 (±8.9)	64.8 (±10.4)	**<0.01**
catecholamine administration-no. (%)	38 (25.3)	51 (40.8)	**<0.01**
**Adverse event**	**no** ***n* = 260**	**yes** ***n* = 15**	
ΔrSO_2_ (Q1–Q3)-%	4 (−1.0–8.0)	2 (−3.5–6.0)	0.32
preoperative rSO_2_ (SD)-%	64.9 (±10.1)	60.9 (±7.2)	0.14
catecholamine administration-no. (%)	20 (7.7)	4 (26.7)	**0.03**
postoperative rSO_2_ (SD)-%	68.4 (±10.1)	62.5 (±8.4)	**0.03**
catecholamine administration-no. (%)	76 (29.2)	13 (86.7)	**<0.01**

Adverse event: in-hospital mortality, conversion to open surgery, CPR. Abbreviations: rSO_2_ = regional cerebral oxygen saturation; SD = standard deviation; POCD = postoperative cognitive disorder; ICU = intensive care unit; CPR = cardiopulmonary resuscitation.= regional cerebral oxygen saturation; Hb = hemoglobin; MAP = mean arterial pressure; CVP = central venous pressure; FiO_2_ = inspirational oxygen fraction; PaO_2_ = arterial partial pressure of oxygen; PaCO_2_ = arterial partial pressure of carbon dioxide; SaO_2_ = arterial oxygen saturation; SpO_2_ = partial oxygen saturation; LVEF = left ventricular ejection fraction; AV = aortic valve; PPG = peak pressure gradient; MPG = mean pressure gradient.

## Data Availability

The data presented in this study are available on request from the corresponding author.

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
