# Peer review of "Cerebral Tissue Oxygen Saturation Is Enhanced in Patients following Transcatheter Aortic Valve Implantation: A Retrospective Study"

_jcm, 2022, doi:10.3390/jcm11071930_

Round 1

Reviewer 1 Report

The manuscript titled: « Cerebral tissue oxygen saturation is enhanced in patients following transcatheter aortic valve implantation: a retrospective study”, by G Schmidt and workers, reports on a retrospective single center study on cerebral tissue oxygen saturation (rSO2) changes following TAVI procedures.

The authors evaluate the variables that may affect the cerebral oximetry comparing them with an univariate and multivariate statistical analysis.

The study is well written and detailed. The statistical analysis is accurate an precise.
However some points are opened to criticism and should be addressed and/or corrected.

In this study the authors don’t take into account the conditions of the cerebral circulation. They don’t refer the presence of associated carotid artery stenosis or cerebrovascular ischemic disease. I believe that it is important to include between the confounding parameters evaluated in the univariate and multivariate analysis the presence or absence of a sever stenosis of the carotid arteries, clinical signs or evidence based on diagnostic tests of an ischemic cerebrovascular disease.

Moreover I believe that is not correct to comparing the patients undergone TAVI procedure under general anesthesia with those under conscious sedation. Patients with general anesthesia and mechanical ventilation have a better management of arterial oxygenation and consequently of the rSO2. Moreover the authors state that patient with an initial higher rSO2 have a less postoperative increase of its value.

In the discussion section the authors talk at length about other considerations. I believe that the discussion should be shortened and focused on evaluation of the change in cerebral tissue oxygen saturation.

Author Response

Dear Reviewer 1,

Thank you for reviewing our manuscript entitled “Cerebral tissue oxygen saturation is enhanced in patients following transcatheter aortic valve implantation: a retrospective study”. Please find attached our point-by-point answers.

We hope your valuable suggestions have been addressed to your satisfaction and you find our manuscript suitable for publication in JCM.

Sincerely yours,

Götz Schmidt

"In this study the authors don’t take into account the conditions of the cerebral circulation. They don’t refer the presence of associated carotid artery stenosis or cerebrovascular ischemic disease. I believe that it is important to include between the confounding parameters evaluated in the univariate and multivariate analysis the presence or absence of a sever stenosis of the carotid arteries, clinical signs or evidence based on diagnostic tests of an ischemic cerebrovascular disease."

  • Carotid artery stenosis was reported as baseline characteristic, and we added prior stroke/TIA (Table 1, L. 106). Overall prevalence was low, and “prior stroke/TIA” comprised of EVERY prior stroke and TIA in the patient’s history (events many years ago, mostly without any residuals). Prior stroke is a common finding in old patients represented by our cohort (mean age 81y), and none of the patients had acute stroke. Same consideration applies for carotid artery stenosis. All TAVI procedures were scheduled as elective procedure, thus, none of the patients had hemodynamic relevant carotid artery stenosis or relevant stroke at the timepoint of the intervention. Otherwise, they would not have been scheduled for TAVI (e.g. endarterectomy first) and would therefore not be included in our study cohort. No patient showed clinical signs of intraprocedural stroke. Thus, we believe stroke is not a relevant confounder in our study, which we could include in the univariate or multivariate analyzes, as its prevalence is overall low, but equal at each timepoint (rSO2 preoperative, rSO2 postoperative): We evaluated the course of paired rSO2 values ensuring intraindividual comparability concerning stroke/tia/carotid artery stenosis is warranted. This position is supported by various studies (including RCT) analyzing rSO2, which reported stroke as a baseline parameter only (Heringlake et al, Anesthesiology, 2011; Sorensen et al, J Clin Monit Comput, 2016) or did not give any statement on the occurrence of prior stroke (Wong et al, BMJ Open, 2020; Murphy et al, Brit J Ana, 2014).

"Moreover I believe that is not correct to comparing the patients undergone TAVI procedure under general anesthesia with those under conscious sedation. Patients with general anesthesia and mechanical ventilation have a better management of arterial oxygenation and consequently of the rSO2. Moreover the authors state that patient with an initial higher rSO2 have a less postoperative increase of its value."

  • Indeed, the anesthetic regime (conscious sedation vs. general anesthesia) must be considered with caution. Inclusion criteria consisted of “patients had to be under stable general anesthesia or conscious sedation that was maintained throughout the procedure ” (L99-101), thus, intraindividual comparability was ensured. It should also be noted that several blood gas analyses were carried out in every patient (regardless of the anesthetic procedure) revealing parameters in physiological ranges. However, in our study cohort, 86.9% (Table 1) received conscious sedation, and only 8.0% (n = 22) were scheduled for general anesthesia. Presuming management of rSO2 during TAVI may be improved, rSO2 should just remain the same (otherwise inclusion criterium as above mentioned would not be met). However, still, in our cohort, rSO2 raised. Thus, we believe the influence of the anesthetic procedure on our results is negligible. Nevertheless, we acknowledged this limitation at LL 304-06, 393-94.

"In the discussion section the authors talk at length about other considerations. I believe that the discussion should be shortened and focused on evaluation of the change in cerebral tissue oxygen saturation."

  • This issue has been addressed in shortening the discussion (removed LL 337-40, 345-349, 353-55, 360-63). From the authors’ point of view, the choice of drugs and herein potential beneficial effect of dexmedetomidine (lower incidence of hypercapnia) must still be mentioned in the discussion focused on the confounder parameters as it represents a relevant aspect of anesthesiology .

Reviewer 2 Report

In the present case report, Dr. Götz Schmidt, et al described the impact of changing cerebral tissue oxygen saturation on TAVI. The rSO2 has demonstrated not only hemodynamic change during rapid pacing but also hyper-perfusion on TAVI. This manuscript would be interesting to readers since the usefulness of rSO2 was well-reviewed in this manuscript.  However, there are some issues to consider:

Major issue

  1. Authors have described the impact of preprocedural rSO2, change of rSO2 drop during rapid pacing, and increasing after the procedure.

In the discussion part, the authors have described that desaturation after RVP was followed by transient hyperemia leading to elevated rSO2. Rapid pacing would become desaturation as well as decrease rSO2 since cerebral flow should be decreased during RVP. What is the mechanism of this phenomenon? Please describe it in more detail.

  1. The hyper-perfusion should consider preprocedural or intraprocedural cerebral infarction since there is a risk of fatal hemorrhage due to hyper-perfusion. Did preprocedural lower rSO2 have chronic cerebral infarction? Please describe more about the relationship between rSO2 and the risk of cerebral complications during TAVI.

Author Response

Dear Reviewer 2,

Thank you for reviewing our manuscript entitled “Cerebral tissue oxygen saturation is enhanced in patients following transcatheter aortic valve implantation: a retrospective study”. Please find attached our point-by-point answers.

We hope your valuable suggestions have been addressed to your satisfaction and you find our manuscript suitable for publication in JCM.

Sincerely yours,

Götz Schmidt

"In the discussion part, the authors have described that desaturation after RVP was followed by transient hyperemia leading to elevated rSO2. Rapid pacing would become desaturation as well as decrease rSO2 since cerebral flow should be decreased during RVP. What is the mechanism of this phenomenon? Please describe it in more detail."

  • Today, the phenomenon of transient hyperemia following RVP during TAVI is described, yet not studied thoroughly. It is unknown, whether this transient hyperemia can be confirmed by other methods (e.g. carotid artery doppler ultrasound). Therefore, the mechanism remains unclear, but it can be hypothesized that the accumulation of acid metabolites (H+, Lactate, CO2) during RVP (= circulatory functional arrest for a few seconds) enhances cerebral perfusion (as CO2/pH is highly correlated with CBF as discussed in our manuscript). Added at LL.278-80. Interestingly, during valve implantation without RVP (=self-expanding valve), this phenomenon does not occur, thus, it does can be attributed to RVP alone. A clinically relevant and persistent hyperperfusion syndrome (as it can be seen after carotid endarterectomy) has not been reported following TAVI. Furthermore, Fanning et al stated that hyperemia was NO sign of stroke in their trial [27].

"The hyper-perfusion should consider preprocedural or intraprocedural cerebral infarction since there is a risk of fatal hemorrhage due to hyper-perfusion. Did preprocedural lower rSO2 have chronic cerebral infarction?"

  • None of our patients had chronic cerebral infarction. We added prior stroke/TIA to the baseline characteristics (Table 1). Overall prevalence was low and comprised of EVERY prior stroke and TIA in the patient’s history (events many years ago, mostly without any residuals). Same consideration applies for carotid artery stenosis. All TAVI procedures were scheduled as elective procedure, thus, none of the patients had hemodynamic relevant carotid stenosis. Otherwise, they would have been scheduled for endarterectomy first (and therefore would not be included in our study cohort). Fanning et al [27] also showed that rSO2 is highly specific, but not sensitive to detect cerebral infraction/stroke after TAVI. Additionally, we investigated the course of rSO2 in our patients – thus, cerebral “anatomic” conditions were the same in one patient (as no one showed clinical signs of stroke after the procedure).

"Please describe more about the relationship between rSO2 and the risk of cerebral complications during TAVI."

  • Actually, today,  evidence in that combined point (rSO2 + complications during TAVI) is very limited. Available data has been shown in the manuscript (lower baseline (although not confirmed in our cohort, [26]) , more severe desaturation predicts delirium, LL. 274-76). Besides rSO2, cerebral complications such as stroke (clinical syndrome, but also evidence of ischemic microlesions in postoperative MRI without symptoms, Fanning et al. [27]) are well-known following TAVI. On the other hand, neurologic injury seems comparable between SAVR and TAVI patients (MRI study, Alassar et al., 2015). Supplementary, ischemic lesion may not only be induced by plaque mobilization, but also by hemodynamic conditions during the procedure (e.g. RVP, where a general drop in rSO2 is obvious). However, it is doubtful whether rSO2 can detect those disturbances/events since only a very limited cerebral area (frontal cortex, approx. 2-3cm depth) is covered by near infrared spectroscopy. Nevertheless, when a significant drop of rSO2 is seen, reasons should be evaluated with particular care

Reviewer 3 Report

The manuscript is well written and presented, even if more attention should be paid to English grammar and structure. The study is well conducted and explores a very appealable topic. In order to achieve more clarity, I suggest uniform aims and scopes in the abstract and the introduction's last paragraph. In addition, the conclusions are different in the abstract and the "Conclusions" session. The obvious result that "Hemoglobin was found as a predictor of altered rSO2" should be defended in the "Discussion" session. I also encourage the Authors to discuss why and how this solely finding should justify the present publication. Finally, interquartile range [IQR] is not a range, but a number. If you wanted to present median and 25th-75th percentiles, please show it as median and Q1, Q3. 

Author Response

Dear Reviewer 3,

Thank you for reviewing our manuscript entitled “Cerebral tissue oxygen saturation is enhanced in patients following transcatheter aortic valve implantation: a retrospective study”. Please find attached our point-by-point answers.

We hope your valuable suggestions have been addressed to your satisfaction and you find our manuscript suitable for publication in JCM.

Sincerely yours,

Götz Schmidt

“The manuscript is well written and presented, even if more attention should be paid to English grammar and structure.”

  • The manuscript has been carefully revised addressing this issue.

“In order to achieve more clarity, I suggest uniform aims and scopes in the abstract and the introduction's last paragraph. In addition, the conclusions are different in the abstract and the "Conclusions" session.”

  • Introduction and Conclusions has been revised addressing this suggestion (LL. 14-16, 25-30)

“The study is well conducted and explores a very appealable topic. The obvious result that "Hemoglobin was found as a predictor of altered rSO2" should be defended in the "Discussion" session. I also encourage the Authors to discuss why and how this solely finding should justify the present publication.”

  • The part concerning „Hb is predictor of altered rSO2“ has been clarified at LL 307 – 310 where another reference was added confirming the general association between Hb and rSO2. Basically, the association is no new finding in our study – referring to basic physiologic considerations. However, it proves the correctness of our multivariate model, where PaCO2 was no significant predictor of altered rSO2. Furthermore, as shown in Table 3, Hb following TAVI was lower compared to its preoperative value, but – nevertheless – rSO2 raised.  The observation on Hb alone does not justify our present publication, but the finding of altered rSO2 following TAVI (and the results of the subgroup analysis), that cannot be attributed to altered hemodynamic parameters alone (e.g. PaCO2 as discussed) does justify the publication.

“Finally, interquartile range [IQR] is not a range, but a number. If you wanted to present median and 25th-75th percentiles, please show it as median and Q1, Q3."

  • Caption “IQR” was replaced by “Q1 – Q3” in the manuscript (Manuscript text and figures)

Round 2

Reviewer 1 Report

my suggestions have been largely satisfied, so I believe your manuscript is suitable for publication

Reviewer 2 Report

This is well revised.

There are no further comments.

Reviewer 3 Report

The manuscript significantly improved after revisions. I have no further comments or edits.